# West-Siberian Chernozem: How Vegetation and Tillage Shape Its Bacteriobiome

**DOI:** 10.3390/microorganisms11102431

**Published:** 2023-09-28

**Authors:** Natalia Naumova, Pavel Barsukov, Olga Baturina, Olga Rusalimova, Marsel Kabilov

**Affiliations:** 1Institute of Soil Science and Agrochemistry, Siberian Branch of the Russian Academy of Sciences, 630090 Novosibirsk, Russia; barsukov@issa-siberia.ru (P.B.); rusalimova@issa-siberia.ru (O.R.); 2Institute of Chemical Biology and Fundamental Medicine, Siberian Branch of the Russian Academy of Sciences, 630090 Novosibirsk, Russia; baturina@niboch.nsc.ru (O.B.); kabilov@niboch.nsc.ru (M.K.)

**Keywords:** 16S rRNA genes, soil bacteria, undisturbed steppe, wheat, conventional tillage, no tillage

## Abstract

Managing soil biodiversity using reduced tillage is a popular approach, yet soil bacteriobiomes in the agroecosystems of Siberia has been scarcely studied, especially as they are related to tillage. We studied bacteriobiomes in Chernozem under natural steppe vegetation and cropped for wheat using conventional or no tillage in a long-term field trial in the Novosibirsk region, Russia, by using the sequence diversity of the V3/V4 region of 16S rRNA genes. *Actinobacteria*, *Acidobacteria*, and *Proteobacteria* summarily accounted for 80% of the total number of sequences, with *Actinobacteria* alone averaging 51%. The vegetation (natural vs. crop) and tillage (ploughed vs. no-till) affected the bacterial relative abundance at all taxonomic levels and many taxa, e.g., hundreds of OTUs. However, such changes did not translate into α-biodiversity changes, i.e., observed and potential OTUs’ richness, Shannon, and Simpson, excepting the slightly higher evenness and equitability in the top 0–5 cm of the undisturbed soil. As for the β-biodiversity, substituting conventional ploughing with no tillage and maintaining the latter for 12 years notably shifted the soil bacteriobiome closer to the one in the undisturbed soil. This study, presenting the first inventory of soil bacteriobiomes under different tillage in the south of West Siberia, underscores the need to investigate the seasonality and longevity aspects of tillage, especially as they are related to crop production.

## 1. Introduction

For quite some time by now, the importance of soil biodiversity in soil quality has been recognized [1,2]. Many agricultural techniques are currently employed to sustain agricultural soils, including managing soil biodiversity by reduced, minimal, or no tillage in attempts to partially reconcile agricultural production and biodiversity. No-till farming has established itself as a technology that cannot be ignored [3], also because lower carbon losses from no-till soil can mitigate the risks associated with global warming: for instance, results obtained on semi-arid lands showed that no-tillage had markedly higher soil organic carbon stocks [4,5].

The conversion of undisturbed steppe areas to cropped land drastically alters the aboveground community, as well as the physiochemical and biological environments. Consequently, such conversion also modifies soil environment for microorganisms, and changes their communities in composition and structure, i.e., diversity. As soil microorganisms are crucial for nutrient cycling, carbon mineralization and other ecosystem processes such as microbial diversity studies, facilitated by advances in sequencing methodology, have been drastically boosted in the last decades, and especially in diverse agrotechnological contexts, including tillage. Notably, even virus abundance and community structure were recently found to be influenced by land use and tillage practices [6].

As for the most functionally diverse component of soil microbiota, i.e., bacteria, so far there is no unequivocal conclusion about whether bacterial biodiversity increases or decreases, if it changes at all, due to reduced or no tillage. For instance, a positive effect of reduced tillage on biodiversity indices, other than richness, was revealed in an extensive study in France [7], whereas the negative impact of the no-till management on soil bacterial diversity was reported for long-term field experiments in the USA [8] and Belgium [9]. Moreover, we did not manage to find studies comparing soil bacterial diversity in cropped fields under different tillage regimes with soil bacterial diversity in the adjacent undisturbed ecosystem to grasp the magnitude of changes between them. Besides, studies of bacteriobiome diversity in arable soils in West Siberia, the important grain-producing region of Russia, are lacking, and nothing is known about soil bacterial taxonomic diversity there in relation to vegetation, tillage, and soil properties. The aim of this study was to reveal the bacteriobiome composition and structure in Chernozem under the condition of natural vegetation or having been cropped for wheat by conventional or no tillage in a long-term field experiment in the Novosibirsk region, Russia, by using 16S rRNA gene diversity.

## 2. Materials and Methods

### 2.1. Experimental Site and Conditions

The area where the field trial was performed was described earlier in our report about soil mycobiomes [10] (https://www.mdpi.com/2075-1729/12/8/1169, accessed on 23 August 2023). Briefly, the experimental field was located in the Novosibirsk region, Russia (54°4′6″ N, 79°36′3″ E) in the forest-steppe zone with a sharply continental climate and Luvic Endocalcic Chernozem (Siltic) [11] veing the most prevalent and agriculturally important soil of the region.

### 2.2. Experimental Setup

The field trial was also described earlier [10]. Briefly, it was started in 2009 on an area of 40 ha when a portion of the conventionally tilled soil (CT, mouldboard ploughing in the fall and disking in the spring) was subjected to the no-till technology (NT); both plots receiving the same rates of herbicides and fertilizers at the same time.

The wheat grain yield, harvested at the beginning of September, 2021, reached 4.8 t ha^−1^ in the NT field and 4.1 t ha^−1^ in the CT field. We also included in the study an undisturbed site (Un), adjacent to the experimental field and occupied by a true bunchgrass steppe (with *Stipa capillata*, *Festuca valesiaca*, some *Poa* spp., and *Puccinella* sp. prevailing) to obtain information about the zonal soil microbiome as a reference for the arable soil.

### 2.3. Soil Sampling and Chemical Analyses

Soil was sampled in October 2021 from 0–5 and 5–15 cm layers in five individual replicates from each layer. In total, 30 soil samples were collected and chemically analyzed as described before [10]. Briefly, soil pH ranged from 6.3 to 6.8, total soil carbon content ranged from 3.6 to 4.2%, and total soil nitrogen content was 0.29–0.37%.

### 2.4. DNA Extraction, Amplification and Sequencing

The DNeasy PowerSoil Kit (Qiagen, Hilden, Germany) was used for total DNA extraction according to the manufacturer’s instructions; the bead-beating was performed with the TissueLyser II (Qiagen, Hilden, Germany) for 10 min at 30 Hz. Agarose gel electrophoresis was used to assess the quality of the extracted DNA; further purification of DNA was not needed. 

The 16S rRNA genes were amplified with the primer pair V3/V4, combined with Illumina adapter sequences [12]. PCR amplification was performed as described earlier [13]. A total of 200 ng PCR product from each sample was pooled together and purified through MinElute Gel Extraction Kit (Qiagen, Hilden, Germany). The obtained amplicon libraries were sequenced with 2 × 300 bp paired-ends reagents on MiSeq (Illumina, San Diego, CA, USA) in SB RAS Genomics Core Facility (ICBFM SB RAS, Novosibirsk, Russia). The read data reported in this study were submitted to the NCBI Short Read Archive under bioproject accession number PRJNA845814.

### 2.5. Bioinformatic Analysis

To analyze the obtained raw sequences, we employed UPARSE pipeline [14] and used Usearch v.11.0.667: the procedure involved length trimming; merging of paired reads and removing less than 350 nt; read quality filtering (-fastq_maxee_rate 0.005); discarding singleton reads; merging of identical reads (dereplication); removing chimeras. The UPARSE-OTU algorithm was used to perform operational taxonomic unit (OTU) clustering, and the clusters were taxonomically attributed by way of SINTAX [14] and 16S RDP training set v.16 [15] as a reference. Then, after eliminating archaeal sequences from the data matrix, we calculated the ratio of the number of taxon-specific sequence reads to the total number of sequence reads, to profile the relative abundance of taxa, expressed as percentage (taxonomic structure) of the obtained bacteribiome assemblages, i.e., a collection of different species at one site at one time [16]. 

The PAST software v. 4.12 [17] helped us to rarefy the obtained OTUs datasets for each field and soil layer: the individual rarefaction graphs (not given here) showed that bacterial OTU numbers plateaued as the number of sequences increased. Therefore, the sampling effort, being near saturation for all samples, allowed us to compare biodiversity [18]. 

### 2.6. Statistical Analyses

Statistical analyses (descriptive statistics, correlation analysis, ANOVA and PCA) were performed by using Statistica v.13.3 a (TIBCO Software Inc., Palo Alto, CA, USA) and PAST [17] software packages. OTUs-based α-biodiversity indices, as well as β-biodiversity (based on Bray-Curtis dissimilarity distance) were calculated using PAST [17]. Factor effects and mean differences in post hoc comparisons by Fisher’s LSD test were considered statistically significant at the *p* ≤ 0.05 level.

## 3. Results

### 3.1. General Taxonomic Diversity

After quality filtering and chimera removal, a total of 4116 different OTUs were identified at a 97% sequence identity level: the overwhelming majority (4100 OTUs) were *Bacteria*, with the rest representing the *Archaea* domain (removed from further analyses). In total, 23 bacterial phyla were found with 87 identified class-level clusters, of which 16 were not explicitly classified.

Most of the total number of bacterial OTUs belonged to the *Proteobacteria* phylum (817, or 20% of the OTU richness), with *Actinobacteria* (695 OTUs) and *Acidobacteria* (593 OTUs) being the second- and third-most OTU-rich phyla, accounting for 17 and 14% of the total number of OTUs, respectively. Notably, however, many of the OTUs (667 OTUs, or 16%) were not identified even to a phylum level. 

As for the relative abundance, the dominance of *Actinobacteria*, *Acidobacteria*, and *Proteobacteria* phyla was much more pronounced: together, they accounted for 77–82% of the total number of sequences (Table 1). *Actinobacteria* was the ultimate dominant phyla in this study, with 55% of the relative abundance in the undisturbed soil and 42 and 48% in the ploughed and no-till soils, respectively. The relative abundance of bacterial sequence reads that could not be assigned below the domain level accounted for 7% of the total number of sequences in the soil bacteriobiome of the experimental fields. *Verrucomicrobia*, *Gemmatimonadete*s, *Chloroflexi*, and *Bacteroidetes* were the moderate dominants, i.e., accounting for 1–5% of the relative abundance in the studied soil samples.

At the class level, the dominance of *Actinobacteria* phyla translated into the dominance of its *Thermoleophilia* and *Actinobacteria* classes (Table 1), whereas the dominance of the *Acidobacteria* phylum mostly resulted from the dominance of its Group_6 and Group_16 classes. The *Proteobacteria* phylum was mainly represented by its *Alphaproteobacteria* class. 

### 3.2. Bacterial Taxonomic Diversity as Related to the Experimental Fields

*Actinobacteria* were 1.3 times more abundant in the 0–5 cm soil with an undisturbed structure and plenty of plant residues, i.e., undisturbed soil and NT treatment as compared with the CT one (Table 1), whereas *Proteobacteria* did not demonstrate any tillage-related differential abundance in both layers. The *Gemmatimonadetes* and *Chloroflexi* were more abundant in the cropped soil as compared with the undisturbed one, where *Verrucomicrobia* prevailed as compared with the cropped fields. As for the class taxonomic level, the CT soil had a markedly higher *Acidobacteria_*Gp16 abundance than the other two soils (Table 1). Contrary to that, *Actinobacteria* class showed a 1.6–1.8 times increased abundance in the 0–5 cm layer of the undisturbed and NT soil, as compared with the CT soil. *Alphaproteobacteria* was 1.6 and 1.3 times more abundant in the bacteriobiome of the undisturbed and NT soils, respectively, than in the CT soil (Table 1). 

The results for the order and family taxonomic levels are given in Appendix A. 

As for the sequence reads clusters at the genus level, 179 genera were affected by tillage and/or the soil layer, with 41 other genera being close to that, i.e., having *p*-values in the 0.5–0.10 range. The representatives of *Acidobacteria*_Gp16 had a higher presence in the CT soil, as compared with the other two soils, in both layers, whereas *Acidobacteria_*Gp6 had an enhanced presence in the 0–5 cm layer. Except for the *Solirubrobacter*, all other dominant bacteriobiome genera were found to have differential abundances between the fields either in the 0–5 or 5–15 cm layers.

Overall, two-way PREMANOVA showed a statistically significant (with *p*-values lower than 0.01) influence of the field and soil layer at all taxonomic levels (Table 2), with the field and layer interaction being statistically significant at all levels below the phylum one. As sources of bacterial taxa abundance variance, the field and soil layer contributed about 1/3 each at the levels from class to OTU; at the phylum level, the field contributed almost half, with the layer contribution being only 12%. 

The location of soil samples in the plane of the first two principal components visualizes very well the relationship between the fields and layers (Figure 1). 

The OTUs, contributing ≥ 1.0% to the total number of sequence reads in a sample in the 0–5 cm soil layer, averaged 15, 14, and 12 OTUs, respectively, in the undisturbed, CT, and NT fields (Figure 2), summarily accounting for 27–29% of the total number of sequence reads. In the 5–15 cm layer, the number of dominant OTUs was 24 in the undisturbed and CT soils, and slightly less (20) in the NT soil, summarily accounting for 48, 42, and 36% of the sequence reads abundance. Over all soil samples, the assemblage of the dominant OTUs embraced 38: thus, the overwhelming majority (99%) of the total number of OTUs was minor or rare bacteriobiome members. 

Some of the prevailing OTUs were specific for the studied experimental fields. The undisturbed soil had as its unique dominants *Bacillus* sp., *Pseudonocardi*a sp., and unclassified representatives of *Spartobacteria*_gis, *Acidobacteria*_Gp4, *Solirubrobacterales*, and *Rhizobiales*. As its specific dominants, conventionally ploughed soil had three OTUs, unclassified to the species level and representing the *Acidobacteria*_Gp16 class. Notably, the no-till soil had no unique dominants at all (Table 3). 

As for the OTUs which were common for the studied fields, there were three in the 0–5 cm layer and 16 in the 5–15 cm (Figure 2a). In the top layer, the common OTUs were *Microlunatus* sp., unclassified *Solirubrobacterales*, and *Rubrobacter* sp., each representing a different class of the *Actinobacteria* phylum (*Actinobacteria*, *Thermoleophilia*, and *Rubrobacteria,* respectively). Several hundreds of OTUs were differentially abundant between the fields (Figure 2b), in both soil layers the biggest difference was between the undisturbed and ploughed fields, and the smallest difference was revealed between the cropped fields.

As for the lower layer, most of its common-for-all-fields OTUs also represented *Actinobacteria* (11, with 5 of the *Thermoleophilia* class); two OTUs represented *Acidobacteria* (Groups 6 and 16), whereas *Gemmatimonadetes* and *Proteobacteria* were represented by one OTU each (unclassified below the phylum level in the case with *Gemmatimonadetes* and below the *Rhizobiales* order). Notably, almost twice as many dominants were found in the bacteriobiome of the lower layer, as compared with the top one (Figure 2a).

### 3.3. Bacteriobiome α- and β-Biodiversity

The α-biodiversity indices did not differ significantly, being practically similar in all studied fields, except for the bacteriobiome evenness and equitability, which were higher in the undisturbed field than in the cropped ones (Table 4).

As for the β-biodiversity, the cropped samples were distant from the undisturbed soil samples under natural vegetation, but the no-till samples were separated from the conventionally ploughed one, despite being located closer to the ploughed field than to the undisturbed one (Figure 3).

## 4. Discussion

Our study provided the first inventory of soil bacteriobiomes in the south of West Siberia, unequivocally showing that undisturbed soil under natural steppe vegetation and wheat-cropped soil under different tillage regimes differed from each other in their bacteriobiome’s composition and structure. 

### 4.1. Soil Bacteriobiome: General Outline

The finding of the *Thermoleophilia* class of the *Actinobacteria* phylum to be ultimately prevailing in the soil of all three fields was somewhat unexpected. The representatives of the class are often found in soils by metagenomics [19,20] and generally seem to prefer environments subjected to relatively low temperatures, a rather high UV influx, and/or low water activity [21]. Since the bacteria are difficult to isolate by conventional laboratory methods, which need to be modified to select slow-growing bacteria (oligotrophic media, extended incubation periods, etc.), not much is known about their physiology and ecological preferences. Yet, like other representatives of the *Actinobacteria* phylum, namely the *Actinobacteria* class, they are likely to be involved in cellulose and hemicellulose decomposition [22]. The high abundance of the *Thermoleophilia* actinomycetes in our study may have been due to the time of soil sampling: the soil was sampled at the end of October, more than a month after the wheat was harvested in the cropped fields in the region, and at the very end of the growing season in the undisturbed plot, with a rather low (+3.5 °C) average monthly temperature in the region, alternating between positive at daytime and negative at nights. Therefore, the environmental conditions with plenty of dead plant material, low temperatures, and enough moisture benefited *Thermoleophilia* in proceeding with plant material decomposition. This dynamics issue is indirectly corroborated by the results of Legrand et al. (2018) [7], who, after studying the effect of tillage on bacteriobiome diversity in soil samples collected in April, i.e., at least at the first trimester of the growing season in France, reported *Actinobacteria* as the third-ranked in relative abundance and OTUs’ richness as 11% and 12%, respectively: compare this with the 42–48% of the *Actinobacteria* sequence relative abundance in the cropped fields in this study (Table 1).

Our Shannon’s α-biodiversity estimate averaged 5.44 over all of the samples studied, being very close as [9,23] or lower [24,25,26] than the estimates obtained in other studies. Our finding that the soil bacteriobiome was equally diverse in all three fields does not fully agree with previously reported results, as there are no unequivocal conclusions concerning the effect of no-till on bacterial α-diversity indices. Some studies concluded that that no-till practices lower soil bacterial diversity [8,27,28], while other researchers found that no-till soil bacteriobiomes had a higher Shannon index [24]; in some reports, the Shannon index in the no-till soil seemed almost exceptionally high as compared with the one in the conventionally tilled soil (9.5 vs. 6.9 [25]). Yet other studies reported no difference in the Shannon index between the soils under conventional and reduced tillage [5,9]. There are also reports that no-till soil (from the longest no-till field experiment in the world) had a higher bacterial richness and five unique phyla [29], and that species richness and evenness were significantly higher in fields under minimum tillage practices in comparison with the fields under conventional tillage [7]. We believe that such ambiguity, alongside environmental, agronomical, and experimental variables, may be caused by the diversity of microbiome research methodology, especially bioinformatic pipelines and taxa clustering.

### 4.2. The Effect of Soil Tillage on Soil Bacteriobiome

Our finding that the no-till soil bacteriobiome was still closer to the conventionally ploughed soil bacteriobiome agrees with the recently published results of an extensive study in Sweden [30], which concluded that no-till soil communities, as compared with the conventional ones, revealed only a slightly higher similarity to abandoned fields and semi-natural grasslands; therefore, their contribution to biodiversity conservation was considered negligible. 

Other studies assessing the impacts of long-term reduced tillage or no-till management on bacterial communities in agricultural soils revealed, by employing the same methodology, i.e., the sequencing of the 16S rRNA gene, that most variability in bacteriobiome composition was observed in its low abundance members [8]. Our finding that the relative abundance of bacterial sequence reads differed between the tillage treatments already at the high taxonomic levels, i.e., the dominant phyla’s relative abundance (Table 1 and Appendix A), and lower taxa as well, which was confirmed by PERMANOVA at all taxonomic levels, clearly did not comply with that result. This discrepancy may be due to the different crop (wheat vs. soybean), the different soil (Chernozem vs. Commerce sandy loam, i.e., not very well-drained Mississippi alluvial soil), the time of the sampling, and other factors. 

Some studies found *Proteobacteria* and *Acidobacteria* phyla to be the most abundant in reduced tillage and no-till soils [8,9,29]. This contrasts with our study where the *Actinobacteria* phylum was found to be ultimately dominating, with its relative abundance increasing from ploughed soil to the no-till soil to the undisturbed soil, accounting in the latter for more than half of the bacteriobiome. There can be little doubt that the *Actinobacteria* dominance in our study was due to plenty of plant residues being present for their decomposition and utilization at least for a month prior to sampling in our study, providing a key ecological niche [31]. 

Like the last three studies, referred to above, in our study, the representatives of the *Acidobacteria* phylum also ranked second in abundance. However, our finding that the phylum was 1.5 times more abundant in the conventionally ploughed soil than in the no-till and undisturbed ones could seem somewhat surprising, as the phylum was recently shown to be prevalent in the undisturbed Chernozem under birch forest in the forest steppe zone of the same region [32], and thus appeared to be associated with the undisturbed humus- and plant-matter-rich soil environment. However, translated to the class level, such phylum abundance did not seem surprising at all, as its higher presence in the ploughed soil was due to the increased presence of its *Acidobacteria_*Gp6 and Gp16 classes, i.e., precisely because of the ploughing, as these classes were shown to increase quite drastically from the topsoil to the underlying layers in the undisturbed Chernozem under birch forest, developed in the same forest-steppe zone on exactly the same parent rock as the Chernozem in our study [32]. The predominance of *Acidobacteria* in the subsoil was also revealed in the grasslands in Germany [33], and the enrichment of oligotrophic *Acidobacteria* under conventional tillage was also reported [7]. Our results showed that the higher prevalence of such dominant phyla as *Gemmatimonadetes* and *Chloroflexi* can be regarded as characteristic for both of the cropped soils studied, and specifically for wheat. 

At the order level, *Geodermatophilales, Gemmatimonadales,* and *Rhodospirillales* were the dominant orders with higher abundance in the cropped soils. The *Geodermatophilales* (*Actinobacteria*/*Actinobacteria*) representatives mainly inhabit bulk and degraded soils but may be associated with plant rhizosphere and even found as plant endophytes. Culture-independent studies revealed their high diversity and frequent predominance in the ecosystems where these actinomycetes are believed to play a key role in biogeochemical cycles [34]. As for the taxa marking the cropped soils with their increased abundance, they were *Actinobacteria* for the no-till soil and *Acidobacteria* (Gp16 and Gp6) for the conventionally tilled soil.

### 4.3. Bacterial Genera and OTUs That Differentially Increased in the Undisturbed Soil

As for the undisturbed soil under natural steppe vegetation, the dominance of *Actinobacteria* and *Acidobacteria* in the bacteriobiome of this soil under natural vegetation with *Stipa krylovii* complied with the phyla dominance in soil under *Stipa bungeana* grassland in the Loess Hilly region in China [35]. The main dominant bacterial phyla with higher abundance than those in both cropped soils were *Verrucomicrobia* with its *Spartobacteria* class and *Acidobacteria*_Gp4. The substantial prevalence of the *Actinobacteria* and *Verrucomicrobia* phyla, as well as the *Acidobacteria*_Gp4 class, can be regarded as characteristic for undisturbed Chernozem under natural vegetation, at least late in the growing season. Our finding that the diversity and composition of soil *Acidobacteria* groups can indicate past land-use change from undisturbed steppe to arable land in general agrees with the results reported by Kim et al. (2021) [36] for forest conversion to farmland in Korea; however, specific patterns of acidobacterial diversity shifts were different between the studies, most likely due to different initial ecosystems (steppe vs. forest), agricultural practices (no manure vs. manure, resulting in the several-times higher organic matter content in the arable soils of the experimental field in [36]), longevity of the field trials, etc. Anyway, the *Acidobacteria* phylum, as one of the major bacteriobiome dominants in soil [37], and Chernozem in particular [38], undoubtedly deserves more research to obtain a better insight into its ecophysiology and behavior in agronomically important contexts. 

### 4.4. General Comments

This study showed that substituting conventional ploughing with no-tillage and maintaining the latter for 12 years brought soil bacteriobiomes that were closer in β-biodiversity to the one in the undisturbed soil under natural steppe vegetation, clearly separating it from the soil bacteriobiome under conventional tillage. This finding agrees with the idea that the physicochemical properties of the soil environment, altered by no-tillage, provide more ecological niches and hence differentiation of the “opportunity space” [9,39] between the fields. Hopefully, over the following years of the experiment, the soil health will be improved [40], while the production performance will be sustained or increased.

The PERMANOVA analysis of the obtained data clearly showed the effect of tillage on the bacteriobiome relative abundance at all taxonomical levels for both soil layers, and ANOVA found hundreds of OTUs that were differentially abundant between the fields. We want to emphasize that here we did not correct for the multiple testing, as the main aim was to show the main trend, i.e., that the list of differentially abundant OTUs between the undisturbed and conventionally ploughed soil embraced twice-as-broad an OTUs spectrum as the one of the OTUs that were differentially abundant between the two similarly cropped, but differently tilled, fields. 

The differences between the fields were revealed mostly for the OTUs that were rather far from notable prevalence; yet, we had expected at least a small effect on such bacteriobiome α-biodiversity indices as Shannon’s or Simpson’s. Our finding that the soil bacteriobiome α-biodiversity indices showed tillage-related differences only in evenness and equitability most likely resulted from much more diverse vegetation in the undisturbed steppe, which provided (a) many different microhabitats for microorganisms and (b) more chemically versatile plant matter input in the soil, thus benefiting a greater number of bacterial species and, consequently, bacteriobiome evenness and equitability. Other α-biodiversity indices (OTUs’ richness, Chao-1, Shannon, Simpson, Dominance, Berger-Parker) failed to differentiate the biodiversity at the field level, despite statistically significant differentiation in some OTUs’ relative abundance. As for the β-biodiversity, estimated by the dissimilarity index based on Bray–Curtis distance, it was more effective at catching differentiation among the fields, clearly separating the studied bacteriobiomes from each other, with the no-till soil being in between the conventionally ploughed and the undisturbed natural soils. 

It is worth emphasizing that including the undisturbed soil into comparative soil metagenomic surveys, as we wrote before [10] (p. 14), “provides a very important reference, crucial for restoring and sustaining soil microbial biodiversity in future”. Other researchers also emphasized that no studies have combined microbiome analyses “with a reference dataset to address to what extent the soil communities of the agricultural soils bear resemblance to more natural habitats” [30] (p. 1023). We found one study of soils under no-till management in Argentina, in which plots under undisturbed natural vegetation were also included [41]. The inclusion of the undisturbed adjacent soil extended the ecological perspective and gave a positive aspect to our study. Therefore, the inclusion of the undisturbed adjacent soil extended the ecological perspective and gave a positive aspect to our study. 

As for bacterial biomarkers that were specific to each tillage treatment, we did not attempt to study temporal dynamics, so the biomarkers we found might be regarded as such only for the same period. Undoubtedly, temporal dynamics should be included in any comprehensive diversity studies for a formal and structured quantification of their variation due to various environmental factors [42], providing better insights into microbial community behavior with their trends, causality, and prediction [43]. Thus, omitting (due to funding and human resources constraints) the longitudinal aspect of soil bacteriobiome in our study can be considered somewhat of a drawback, albeit the study making an interesting North Asian contribution into the global soil bacteriobiome data set. 

Since rare sequences are hardly likely to seriously affect the ecological context [44], here, we presented results mainly for the dominant taxa, despite the vast sets of OTU clusters that were differentially abundant in the studied soils: often there is very limited, if any, ecophysiological knowledge pertinent even to the dominant OTUs, not to mention the minor or rare ones [45]. Thus, ecologically interpret OTU/species assemblages assessed by analysing environmental DNA less speculative and more factually justified, much research is yet necessary for improving the ecological annotation in the relevant reference databases.

To assess bacterial taxonomic diversity, we used one of the variable regions of the 16S rRNA gene, as such analysis has been, so far, the most common tool in bacterial taxonomic studies. However, the gene can have multiple copies within a genome [46,47], and hence, the presence of such multiple heterogeneous 16S rRNA gene copies may, and very likely does, over-estimate community biodiversity [48,49]. Our finding that the phyla *Proteobacteria* and *Firmicutes* were not prominent in the studied soils, together with the highest amount of variation in the copies of their 16S rRNA genes [50,51], suggests that the contribution of these phyla in the over-estimation of biodiversity, if any over-estimation occurred, was not at least maximal. Also, such phyla as *Actinobacteria* and *Acidobacteria*, ultimately dominating in our study and showing less variation in the copies of their 16S rRNA genes [50] implies low biodiversity over-estimation because of the phyla. Overall, there is a certain ambiguity concerning the effect of intragenomic sequence heterogeneity among multiple 16S rRNA genes, which may affect species classification [52], or may be unlikely to have a strong effect on the classification of taxa [53] due to the limited heterogeneity in the copies of the 16S rRNA genes [50].

At the same time, as variable regions of the 16S rRNA gene are unlikely to ever adequately discriminate between species [48], the true species richness of soil bacteriobiomes in our study may be underestimated. On the other hand, as discussed above, the intragenomic variation in the 16S gene can overestimate true bacterial diversity. However, as Werner Heisenberg put it, “In science … the object of research is no longer nature in itself, but rather nature exposed to man’s questioning…” [54] (p. 105). True values of species richness of soil microbiota are hardly ever attainable. Anyway, the fact that soil bacteriome diversity data, obtained here by sequencing amplicons of 16S rRNA genes, resulted in ecologically relevant and meaningful bacteriobiome patterns in the context of our study, proves that the methodology grasps a substantial portion of true bacterial diversity in soil.

The biodiversity indices, currently employed in microbiome studies as calculated on the basis of the number of nucleotide sequences reads in a study, provide a uniform method to describe and compare the biodiversity of different biomes, ecosystems, areas, and habitats. However, α-biodiversity indices, calculated with such data, do not always seem ecologically sensitive, probably due to the presence of multiple copies of the gene within a genome. Yet, recently, it was concluded that 16S rRNA gene copy number normalization (in order to bring sequence-derived biodiversity estimates closer to the population-derived ones) does not provide more reliable conclusions in metataxonomic surveys [55]. 

## 5. Conclusions

Here, we presented the first survey of the bacteriobiome diversity in Chernozem under different vegetation (undisturbed steppe vs. wheat) and tillage (conventional vs. no tillage) treatments. We found a clear effect of the tillage mode on the relative abundance of some taxa already at the high taxonomic levels, with three studied fields, i.e., natural steppe and wheat-cropped by conventional or no tillage, differing from each other. After 12 years of continuous no-tillage management, the soil bacteriobiome β-diversity differed among the fields, shifting the no-till one notably towards the undisturbed steppe, yet leaving it still closer to the bacteriobiome in conventionally ploughed soil. Further studies, focusing on the longitudal aspect of soil bacteriobiome variability, both seasonal and long-term, can be helpful in finding the main drivers shaping soil microbial diversity and its relationship with crop production. 

## Figures and Tables

**Figure 1 microorganisms-11-02431-f001:**
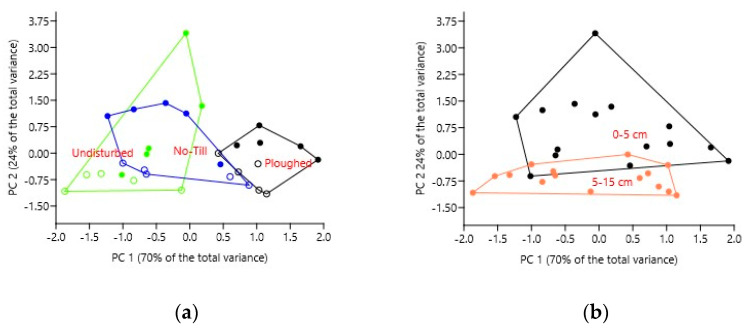
Principal component analysis (based on covariance) of the data matrix with soil samples as rows and bacterial phyla relative abundance as variables for analysis: location of soil samples in the plane of principal components 1 and 2 with convex hulls grouping either fields (**a**) or layers (**b**). Solid circles denote 0–5 cm layer, and open circles denote 5–15 cm layer.

**Figure 2 microorganisms-11-02431-f002:**
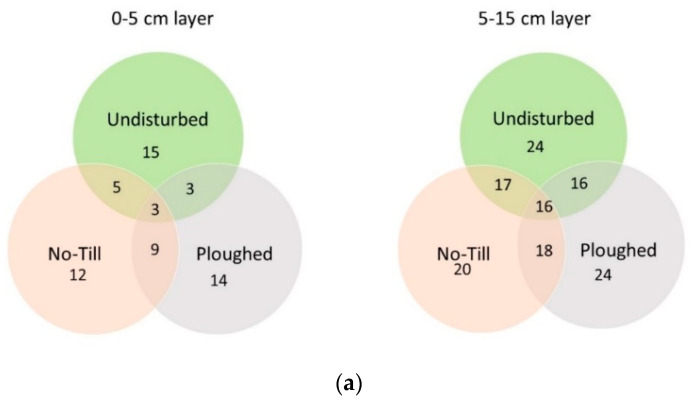
Venns diagram of (**a**) the number of the dominant bacterial OTUs in soil under different tillage treatments with the number of common OTUs (OTUs were considered dominant if they accounted for ≥1% of the total number of sequence reads), and (**b**) of the total number of OTUs and the number of differentially abundant OTUs (at *p* ≤ 0.05 level, Fisher’s LSD test).

**Figure 3 microorganisms-11-02431-f003:**
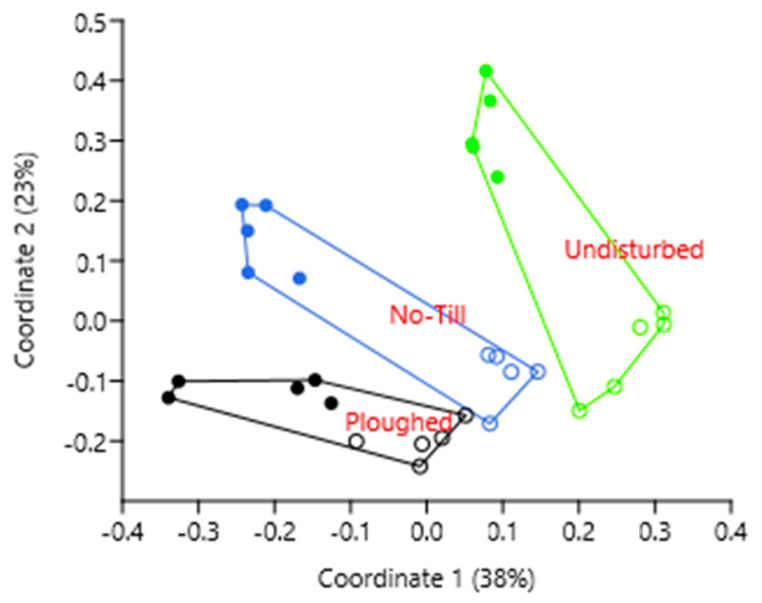
Principal coordinates analysis of the soil bacteriobiome composition (genus level, Bray-Curtis dissimilarity distance) under different soil tillage management in the forest-steppe zone in West Siberia: location of samples in the plane of the first two coordinates. Symbols: solid circles denote 0–5 cm layer, open circles denote 5–15 cm layer.

**Table 1 microorganisms-11-02431-t001:** Relative abundance (%, mean) of the dominant bacterial phyla and classes in Chernozem of the experimental fields in the south of West Siberia.

Taxon	Undisturbed		Ploughed		No Till	
	0–5 cm	5–15 cm	0–5 cm	5–15 cm	0–5 cm	5–15 cm
Phylum level						
*Actinobacteria*	48.2 ^b 1^	58.1 ^d^	38.0 ^a^	43.5 ^ab^	47.5 ^bc^	50.5 ^c^
*Acidobacteria*	19.6 ^a^	18.6 ^a^	32.9 ^b^	32.4 ^b^	18.7 ^a^	24.6 ^a^
*Proteobacteria*	12.7 ^b^	6.0 ^a^	9.3 ^ab^	5.9 ^a^	12.2 ^b^	6.3 ^a^
*Gemmatimonadetes*	1.9 ^a^	2.3 ^ab^	4.4 ^c^	4.3 ^c^	6.6 ^d^	3.5 ^bc^
*Chloroflexi*	1.3 ^a^	1.1 ^a^	3.2 ^d^	2.5 ^bc^	2.8 ^cd^	2.1 ^b^
*Verrucomicrobia*	4.9 ^c^	3.7 ^bc^	1.3 ^a^	1.3 ^a^	1.0 ^a^	1.8 ^ab^
*Bacteroidetes*	1.6 ^c^	0.3 ^a^	0.8 ^b^	0.4 ^a^	1.6 ^c^	0.5 ^ab^
Class level						
*Thermoleophilia*	14.2 ^a^	29.1 ^c^	13.6 ^a^	20.4 ^b^	11.5 ^a^	20.6 ^b^
*Acidobacteria* Gp6	10.9 ^ab^	7.1 ^a^	12.4 ^b^	14.8 ^b^	7.3 ^a^	13.6 ^b^
*Acidobacteria* Gp16	2.1 ^a^	5.5 ^b^	12.4 ^c^	10.4 ^c^	4.6 ^ab^	4.3 ^ab^
*Actinobacteria*	22.0 ^c^	12.5 ^a^	14.0 ^b^	10.1 ^a^	24.5 ^c^	13.9 ^b^
*Alphaproteobacteria*	8.5 ^c^	5.0 ^ab^	5.4 ^ab^	4.0 ^a^	7.2 ^b^	4.6 ^ab^
*Spartobacteria*	4.6 ^b^	3.5 ^b^	1.0 ^a^	1.1 ^a^	0.7 ^a^	1.6 ^b^
*Rubrobacteria*	3.7 ^abc^	2.2 ^ab^	6.8 ^c^	4.4 ^b^	5.7 ^c^	3.1 ^ab^
*Acidobacteria*	0.4 ^a^	0.3 ^a^	3.1 ^cd^	2.8 ^c^	2.1 ^bc^	1.6 ^b^
*Gemmatimonadetes*	1.8 ^b^	0.6 ^a^	3.0 ^c^	1.8 ^b^	4.4 ^d^	1.6 ^b^
*Acidimicrobiia*	2.6 ^ab^	2.4 ^ab^	2.0 ^a^	2.4 ^ab^	2.0 ^a^	3.0 ^b^
*Acidobacteria* Gp4	2.7 ^c^	2.9 ^c^	0.2 ^a^	0.4 ^a^	0.4 ^a^	1.5 ^b^
*Bacilli*	1.3 ^ab^	1.6 ^b^	0.2 ^a^	0.3 ^a^	0.5 ^a^	0.7 ^ab^
*Betaproteobacteria*	1.3 ^b^	0.3 ^a^	1.1 ^b^	0.5 ^a^	1.3 ^b^	0.4 ^a^
*Deltaproteobacteria*	2.2 ^d^	0.4 ^a^	1.7 ^c^	1.0 ^b^	2.4 ^d^	0.8 ^ab^
*Gammaproteobacteria*	0.7 ^b^	0.1 ^a^	0.9 ^c^	0.2 ^a^	1.2 ^c^	0.4 ^ab^
*Chitinophagia*	1.0 ^b^	0.2 ^a^	0.5 ^a^	0.3 ^a^	1.1 ^b^	0.4 ^a^

^1^ Different letters in rows indicate that the values are different (*p* ≤ 0.05, Fisher’s LSD test).

**Table 2 microorganisms-11-02431-t002:** Two-way PERMANOVA results for bacterial taxa.

Source	Sum of Squares	% In Variance	D.f.	Mean Square	F	*p*
	Phylum level					
Field	1979.2	47	2	989.62	15.839	0.0001
Layer	543.91	13	1	543.91	8.7055	0.0018
Interaction	180.1	4	2	90.051	1.4413	0.2215
Residual	1499.5	36	24	62.479		
Total	4202.7	100	29			
	Class level					
Field	1271.2	27	2	635.61	10.515	0.0001
Layer	1689.8	35	1	1689.8	27.956	0.0001
Interaction	357.29	7	2	178.64	2.9554	0.0091
Residual	1450.7	30	24	60.445		
Total	4769	100	29			
	Order level					
Field	1295.3	31	2	647.67	11.778	0.0001
Layer	1185.7	28	1	1185.7	21.562	0.0001
Interaction	360.25	9	2	180.12	3.2755	0.002
Residual	1319.8	32	24	54.992		
Total	4161.1	100	29			
	Family level					
Field	1285.6	31	2	642.82	12.224	0.0001
Layer	1190	29	1	1190	22.631	0.0001
Interaction	362.79	9	2	181.4	3.4496	0.0012
Residual	1262.1	31	24	52.586		
Total	4100.5	100	29			
	Genus level					
Field	1270.4	32	2	635.2	12.198	0.0001
Layer	1112.6	28	1	1112.6	21.365	0.0001
Interaction	331.84	8	2	165.92	3.1862	0.0028
Residual	1249.8	32	24	52.073		
Total	3964.6	100	29			
	OTUs level					
Field	433.33	25	2	216.66	6.3671	0.0001
Layer	385.68	22	1	385.68	11.334	0.0001
Interaction	123.89	7	2	61.944	1.8203	0.0386
Residual	816.69	46	24	34.029		
Total	1759.6	100	29			

**Table 3 microorganisms-11-02431-t003:** Relative abundance (%, mean) of the dominant bacterial genera in Chernozem of the experimental fields in the south of West Siberia.

Genus	Undisturbed		Ploughed		No Till	
	0–5 cm	5–15 cm	0–5 cm	5–15 cm	0–5 cm	5–15 cm
*Actinoplanes*	2.6 ^d, 1^	0.0 ^a^	0.4 ^b^	0.0 ^a^	1.5 ^c^	0.1 ^ab^
*Bacillus*	1.0 ^b^	1.1 ^b^	0.1 ^a^	0.0 ^a^	0.2 ^ab^	0.3 ^ab^
*Blastococcus*	0.0 ^a^	0.0 ^a^	3.3 ^c^	1.4 ^b^	4.7 ^d^	1.5 ^b^
*Gaiella*	1.5 ^a^	13.9 ^d^	6.2 ^b^	11.7 ^c^	2.9 ^a^	10.2 ^c^
*Acidobacteria_*Gp4	2.9 ^c^	2.9 ^c^	0.4 ^a^	0.5 ^a^	0.5 ^a^	1.6 ^b^
*Acidobacteria_*Gp6	11.0 ^ab^	7.1 ^a^	12.4 ^b^	14.9 ^b^	7.3 ^a^	13.6 ^b^
*Acidobacteria_*Gp16	2.1 ^a^	5.6 ^b^	13.0 ^c^	10.9 ^c^	4.7 ^ab^	4.4 ^ab^
*Kribbella*	1.0 ^b^	1.0 ^b^	0.4 ^a^	0.5 ^ab^	0.9 ^b^	0.8 ^ab^
*Microlunatus*	2.5 ^a^	4.1 ^ab^	1.9 ^a^	2.4 ^a^	5.1 ^b^	4.1 ^ab^
*Pseudonocardia*	3.2 ^c^	0.3 ^a^	0.7 ^a^	0.6 ^a^	2.0 ^b^	0.8 ^a^
*Rubrobacter*	3.7 ^ab^	2.2 ^a^	6.8 ^b^	4.4 ^ab^	5.7 ^b^	3.1 ^a^
*Solirubrobacter*	0.8 ^a^	1.5 ^a^	1.4 ^a^	1.3 ^a^	1.8 ^a^	1.0 ^a^
*Spartobacteria_gis*	4.6 ^b^	3.5 ^b^	0.9 ^a^	1.1 ^a^	0.7 ^a^	1.5 ^a^
un. ^2^ *Acidimicrobiales*	0.9 ^b^	1.4 ^c^	0.4 ^a^	1.0 ^b^	0.4 ^a^	1.4 ^c^
un. *Acidobacteria*	0.8 ^a^	0.7 ^a^	3.7 ^c^	3.4 ^bc^	3.1 ^b^	2.3 ^b^
un. *Actinobacteria*	9.2 ^c^	16.7 ^e^	3.0 ^a^	9.0 ^c^	5.5 ^b^	13.1 ^d^
un. *Chloroflexi*	0.4 ^a^	0.5 ^ab^	1.5 ^d^	1.0 ^c^	1.3 ^d^	0.7 ^b^
un. *Gemmatimonadaceae*	1.4 ^b^	0.5 ^a^	2.7 ^c^	1.7 ^b^	3.7 ^d^	1.5 ^b^
un. *Gemmatimonadetes*	0.1 ^a^	1.7 ^b^	1.3 ^ab^	2.5 ^b^	2.1 ^b^	1.9 ^b^
un. *Micromonosporaceae*	2.3 ^c^	0.9 ^b^	0.4 ^a^	0.5 ^ab^	0.6 ^ab^	0.8 ^b^
un. *Rhizobiales*	4.3 ^b^	4.2 ^b^	1.5 ^a^	1.8 ^a^	0.9 ^a^	2.3 ^a^
un. *Solirubrobacterales*	9.1 ^c^	8.9 ^c^	4.8 ^a^	5.5 ^ab^	5.8 ^ab^	7.0 ^b^
un. *Thermoleophilia*	2.6 ^c^	4.6 ^d^	1.0 ^a^	1.8 ^b^	0.8 ^a^	2.3 ^bc^

^1^ Different letters in rows indicate that the values are different (*p* ≤ 0.05, Fisher’s LSD test). ^2^ un. stands for unclassified.

**Table 4 microorganisms-11-02431-t004:** Alpha-biodiversity indices (calculated on the OTU’s basis) of bacteriobiomes in the Chernozem of the experimental fields in the south of West Siberia.

Index	Undisturbed		Ploughed		No Till	
	0–5 cm	5–15 cm	0–5 cm	5–15 cm	0–5 cm	5–15 cm
OTU richness	932 ^ab, 1^	846 ^a^	1158 ^b^	981 ^ab^	970 ^ab^	1019 ^ab^
Chao-1	1349 ^ab^	1160 ^a^	1562 ^b^	1415 ^ab^	1402 ^ab^	1399 ^ab^
Simpson (S)	0.989 ^b^	0.985 ^a^	0.986 ^ab^	0.988 ^ab^	0.987 ^ab^	0.988 ^ab^
Shannon’s	5.55 ^b^	5.11 ^a^	5.51 ^b^	5.32 ^ab^	5.47 ^b^	5.40 ^b^
Evenness	0.28 ^c^	0.20 ^a^	0.22 ^ab^	0.21 ^a^	0.25 ^b^	0.22 ^ab^
Equitability	0.81 ^d^	0.76 ^a^	0.79 ^bc^	0.77 ^ab^	0.80 ^c^	0.78 ^b^
Berger-Parker	0.05 ^a^	0.05 ^a^	0.07 ^b^	0.05 ^ab^	0.06 ^ab^	0.05 ^a^
Dominance (1-S)	0.011 ^a^	0.015 ^b^	0.014 ^ab^	0.012 ^ab^	0.013 ^ab^	0.012 ^ab^

^1^ Different letters in rows indicate that the values are different (*p* ≤ 0.05, Fisher’s LSD); the absence of letters after the values in a row indicates that there was no difference.

## Data Availability

The read data reported in this study were submitted to the GenBank under the study accession PRJNA845814 (https://www.ncbi.nlm.nih.gov/bioproject/ accessed on 23 August 2023).

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
