# Peer review of "West-Siberian Chernozem: How Vegetation and Tillage Shape Its Bacteriobiome"

_microorganisms, 2023, doi:10.3390/microorganisms11102431_

Round 1
Reviewer 1 Report
See attached

Author Response
Thank you very much for reviewing our manuscript and comments for improving it!
|
# |
Reviewer’s comments |
|
Point 1 |
We suggest that the authors remove the parts highlighted in yellow from the document… Line 63 to 64 The area where the field trial was performed was described earlier in our report about soil mycobiome [10] (https://www.mdpi.com/2075-1729/12/8/1169). |
|
Response 1 |
Removed. |
|
Point 2 |
Line 69 to 70 The field trail was also described earlier [10] (https://www.mdpi.com/2075- 69 1729/12/8/1169). |
|
Response 2 |
Removed. |
|
Point 3 |
Line 82 to 83 In total 30 soil samples were collected and chemically analyzed as described before [10] [Error! Bookmark not defined.] https://www.mdpi.com/2075- 82 1729/12/8/1169). |
|
Response 3 |
Removed. |
|
Point 4 |
Line 105 to 106 The OTU sequences were assigned a taxonomy using the SINTAX [Error! Bookmark not defined.] and … |
|
Response 4 |
Removed and corrected. |
|
Point 5 |
Line 249 to 250 This dynamics issue is indirectly corroborated by the results of Legrand et al. (2018) [5], who, after studying the effect of tillage on bacteriobiome diversity. Legrand et al. (2018) [5] or Legrand et al. (2018) [7]??? There is no harmonisation with what is presented in the reference section. |
|
Response 5 |
The reference checked and corrected. |
|
Point 6 |
Line 376 to 377 It is worthwhile emphasizing that including the undisturbed soil into comparative soil metagenomic surveys, as we wrote before [10, p.14], |
|
Response 6 |
We are afraid we cannot remove the indication of the page; but corrected the format according to the MDPI rules. (In the text, reference numbers should be placed in square brackets [ ], and placed before the punctuation; for example [1], [1–3] or [1,3]. For embedded citations in the text with pagination, use both parentheses and brackets to indicate the reference number and page numbers; for example [5] (p. 10). or [6] (pp. 101–105).) |
|
Point 7 |
Line 378 to 379 Other researchers also emphasized that no studies have combined microbiome analyses “with a reference dataset to address to what extent the soil communities of the agricultural soils bear resemblance to more natural habitats” [29, p.1023]. |
|
Response 7 |
We are afraid we cannot remove the indication of the page; but corrected the format according to the MDPI rules. |
|
Point 8 |
Line 412 to 414 Also, as such phyla as Actinobacteria and Acidobacteria, ultimately dominating in our study, showed less variation in the copies of their 16S rRNA genes [Error! Bookmark not defined.], |
|
Response 8 |
Removed and corrected. |
|
Point 9 |
Line 422 to 424 But, as Werner Heisenberg put it, “What we observe is not nature itself, but nature exposed to our method of questioning” (cited from Bokulich et al., 2020 [54, p.4049]), true values of species richness of soil microbiota are hardly ever attainable. |
|
Response 9 |
We changed it into direct citation of the W.H. paper (1958) in the correct format. |

Reviewer 2 Report
Manuscript ID: microorganisms-2603396
Title: West-Siberian Chernozem: how vegetation and tillage shape its bacteriobiome
The manuscript describes the sequencing and analysis of the bacterial community structure in Chernozem samples covered by native vegetation, as well as wheat crop soils with differential management. Overall Actinobacteria, Acidobacteria, and Proteobacteria were the most representative bacterial phyla. The manuscript brings novel information related to the bacterial community conformation in West-Siberian Chernozem area. The manuscript is well redacted and the topic could be of interest to the Microorganisms readers.
I consider the manuscript suitable for publication after addressing some minor corrections
Commentaries:
Line 82, review “[Error! Bookmark not defined.]” some information or symbol is missing
Lines 98-99, add a space to separate paragraphs
Line 106, review “[Error! Bookmark not defined.]” some information or symbol is missing
Line 121, review format in the term “post-hoc” must be in italics
Line 136, review format in “77-82 %”, maybe could be better “77-82%”
Table 1, review, to use “cm” in each column could be repetitively
Line 163, “P” could be “p” and use italics
Line 171, use italics for “p”
Figure 1, improve quality of the figure is some blurred
Line 196, the table head must be on the same page as the table
Figure 3, improve quality of the figure is some blurred
Table 3, review, to use “cm” in each column could be repetitively
Figure 2, in figure description, use italics for “p”
Table 4, review, to use “cm” in each column could be repetitively
Lines 312-314, join this information with the previous paragraph
Lines 322-324, join this information with the previous paragraph
Lines 376-384, review the format in references [10] and [29]
Lines 385-386, join this information with the previous paragraph
Line 414, review “[Error! Bookmark not defined.]” some information or symbol is missing
Line 421-424, use direct references
Author Response
We are awfully grateful to you for your detailed and thorough review helping to improve our manuscript and approving it for publication!
|
# |
Reviewer’s comments and Authors’ responses (changes in the manuscript are highlighted in green)
|
|
Point 1 |
Lines 82, Line 106, 414 review “[Error! Bookmark not defined.]” some information or symbol is missing |
|
Response 1 |
Corrected. |
|
Point 2 |
Lines 98-99, add a space to separate paragraphs |
|
Response 2 |
Corrected. |
|
Point 3 |
Line 121, review format in the term “post-hoc” must be in italics |
|
Response 3 |
Corrected. |
|
Point 4 |
Line 136, review format in “77-82 %”, maybe could be better “77-82%” |
|
Response 4 |
Yes, of course; corrected. |
|
Point 5 |
Tables 1, 3, 4, review, to use “cm” in each column could be repetitively |
|
Response 5 |
It is, yes. But we could not come up with a better way to avoid it, therefore decided to leave the Tables as they are. |
|
Point 6 |
Line 163, 171, “P” could be “p” and use italics |
|
Response 6 |
Corrected. |
|
Point 7 |
Figures 1, 3, improve quality of the figure is some blurred |
|
Response 7 |
It might be so, but since the graph is produced by statistics software (PAST) we cannot do much changing the quality. However, MDPI publishing team usually had no problems with the quality of our PAST graphs. |
|
Point 8 |
Line 196, the table head must be on the same page as the table |
|
Response 8 |
Rearranged properly. |
|
Point 9 |
Figure 2, in figure description, use italics for “p” |
|
Response 9 |
Corrected. |
|
Point 10 |
Lines 312-314, join this information with the previous paragraph |
|
Response 10 |
Joined. |
|
Point 11 |
Lines 322-324, join this information with the previous paragraph |
|
Response 11 |
Joined. |
|
Point 12 |
Lines 376-384, review the format in references [10] and [29] |
|
Response 12 |
Reviewed. |
|
Point 13 |
Lines 385-386, join this information with the previous paragraph |
|
Response 13 |
Joined. |
|
Point 14 |
Line 421-424, use direct references |
|
Response 14 |
So done. And thank you very much for prompting us to do it. |

Reviewer 3 Report
The article is an interesting study of the bacteriobiome of chernozem in Western Siberia under conditions of natural steppe vegetation and wheat cultivation with traditional and no-till farming.
The introduction contains sufficient background information and relevant links.
The study is original because the bacteriobioms of Western Siberia have not been studied before; their dependence on processing methods has also not been studied before.
The relevance is well founded, due to the widespread use of the no-till farming method. The study of microbial diversity and its conservation for agricultural soils is important.
The research design appropriate corresponds to the set goals, is properly technically developed and justified. The analyses are carried out in accordance with the technical standards. The methods, tools, and software are described in sufficient details.
The study contributes to the development of agricultural microbiology, because it is the first inventory of the soil bacteriobiome in Western Siberia with various methods of tillage. Research carried out by the authors seems to be important to the development and enhancement of existing information on this subject.
However, some questions need clarification:
1. What is the reason for the choice of the presented alpha-biodiversity indices (OTU richness, Chao-1, Shannon, Simpson, Dominance, Berger-Parker, Evenness)? There is no explanation of the rationale for the use of these indexes and their interpretations.
2. The study of the zonal soil microbiome in this research is important for understanding the bacterial diversity of undisturbed Chernozem under natural West-Siberian steppe. However, can the use of the zonal soil microbiome as a reference for arable soil be fully applicable? Since the natural soil develops under the conditions of a certain plant community, and the soil under a monoculture does not provide the same species diversity of microorganisms. Consequently, the similarity of the bacteriobiome in conditions of natural steppe vegetation and under wheat crops can hardly be achieved even by no-till farming.
The article can be accepted after minor revision (corrections to minor methodological errors and text editing)
Author Response
Thank you very much for reviewing our manuscript!
|
# |
Reviewer’s comments |
|
|
Point 1 |
What is the reason for the choice of the presented alpha-biodiversity indices (OTU richness, Chao-1, Shannon, Simpson, Dominance, Berger-Parker, Evenness)? There is no explanation of the rationale for the use of these indexes and their interpretations. |
|
|
Response 1 |
The reasons are simple. Often, when we are looking for such indices in other research reports to compare and discuss those we obtained, we find one or two different estimates (most often than not without any justification), and could not use them to our end, only referring to authors’ conclusion about biodiversity changes they revealed. Therefore, since we are aware of such situation, i.e. that other scientists often are looking for the values, we tend to provide as many as we can reasonably get with such authoritative software as PAST (the more so that with electronic media space is not an issue in contrast to the printed media). In our manuscript we provide the reference to the PAST; brief description of the indices and their formulas are given in the freely available PAST manual. Besides, we are not sure that more detailed discussion of the indices should be given within the subject of our manuscript. |
|
|
Point 2 |
The study of the zonal soil microbiome in this research is important for understanding the bacterial diversity of undisturbed Chernozem under natural West-Siberian steppe. However, can the use of the zonal soil microbiome as a reference for arable soil be fully applicable? Since the natural soil develops under the conditions of a certain plant community, and the soil under a monoculture does not provide the same species diversity of microorganisms. Consequently, the similarity of the bacteriobiome in conditions of natural steppe vegetation and under wheat crops can hardly be achieved even by no-till farming. |
|
|
Response 2 |
You are absolutely right that the natural soil develops under the conditions of a certain plant community, and the soil under a monoculture does not provide the same species diversity of microorganisms. And that was exactly one of the aims of the study, i.e. to see how different may be the microbial communities in the same pedo-environmental conditions but under different vegetation and soil tillage. We write about the natural community as a reference not in the sense that it is a standard to strive restoring, but as a mark for comparison to try to infer some ecological meaning into what species changed and why. We totally agree with you about similarity as well, and never wrote that no-till soil should become similar to the undisturbed one. |
|

Reviewer 4 Report
The manuscript with the title "West-Siberian Chernozem: how vegetation and tillage shape its bacteriobiome" is devoted to the urgent problem of biodiversity soil bacteriobiome in the agroecosystems. The study, presenting the first inventory of soil bacteriobiome under different tillage in the south of West Siberia. It is now becoming obvious that successful crop production requires fundamental knowledge about changes in the soil microbiome, taking into account both seasonal and agrotechnical factors. Therefore, in general, I can say that the manuscript corresponds to the main aims and scopes of the journal "Microorganisms".
1. The introduction seems too short to me. The authors should dwell in more detail on the results of works 7-8 and briefly describe the features of soils and their microbiome in Siberia based on data obtained earlier in work 10 and other researchers.
2. Paragraph 2.1. it is necessary to expand and describe the sites from where the samples were taken, specifying the time and conditions of sampling. It may be worth doing this in the form of a map.
3. Were there only 3 sampling points? This is not enough for global conclusions. Were the repetitions made from the same sample or from different ones?
It would be interesting to take a sample from a deeper point where there is no plowing.
4. L82 106 414 [Error! Bookmark not defined.] Please correct.
5. In the discussion, I would like the authors to note whether there are fundamental differences in biodiversity between soils from West Siberian chernozem and chernozems from other regions.
6. Have the elemental composition of soils been analyzed? It would be interesting to see if there is a correlation between the diversity of microbiomes (especially the contribution of actinomycetes) and the presence of biophilic elements and trace elements.
Author Response
Thank you very much for your thorough review and comments to improve our manuscript!
|
# |
Reviewer’s comments
|
|
Point 1 |
The introduction seems too short to me. The authors should dwell in more detail on the results of works 7-8 and briefly describe the features of soils and their microbiome in Siberia based on data obtained earlier in work 10 and other researchers. |
|
Response 1 |
Since the manuscript has been reviewed, besides you, by three other researchers, who did not comment on the Introduction, we are afraid to expand it in order not to confuse them and/or bring more comments. Moreover, although we believe in brevity, applying it to describing the features of soils in Siberia would be a Herculean task, well beyond the aim of the paper; and as for soil microbiome, the studies are too few, and mostly ours. As for works 7-8, we refer to them in Discussion as well and in more details. |
|
Point 2 |
Paragraph 2.1. it is necessary to expand and describe the sites from where the samples were taken, specifying the time and conditions of sampling. It may be worth doing this in the form of a map. |
|
Response 2 |
We agree with you that the text should be expanded, i.e. described in this paper exactly in the same way as in our previous paper (https://www.mdpi.com/2075-1729/12/8/1169) based on the same soil samples and the same DNA extracts, but amplified with ITS primers, because – you are right! – the lack of the info about sites etc. is counter-productive for understanding the content. However, in our attempt to shorten and rewrite the text in 2.1 in order to avoid to be kicked out automatically by the MDPI similarity-checking software (which MDPI ultimately relies on without any human effort involved) because of our previous paper, we accidentally deleted the geographical coordinates, never noticing it. We inserted a sentence with coordinates for any reader to find the field on the map. The time of sampling was indicated and the number of individual soil samples were given in lines 80-81 of the reviewed paper. To emphasize the point about the similarity-checking MDPI soft, let us tell you that for this manuscript you kindly reviewed, the associate editor wrote that “we kindly suggest you to re-write the paragraph, section 2.5 Bioinformatic Analysis, which is similar with the published paper”. We believe that this M&M descriptions, coming from the same team of authors, the same soil samples and the same bioinformatic pipeline (except for the fungi-related part) should be absolutely the same in both papers in order not to introduce any confusion concerning such important parts of research reports!
|
|
Point 3 |
Were there only 3 sampling points? This is not enough for global conclusions. Were the repetitions made from the same sample or from different ones? It would be interesting to take a sample from a deeper point where there is no plowing.
|
|
Response 3 |
From each field we collected five individual soil samples from 0-5 and 5-15 cm layers; overall 30 samples which were individually analyzed: the information was supplied in M&M Section. As you may have noticed, we are far from making global conclusion, emphasizing the effect of the date.
We agree that it would be interesting to take a sample from a deeper layer. But with much less microbial biomass there and much less diversity (like we showed it for the Phaeozem genetic horizons under the undisturbed birch forest at the same latitude but situated more westward in comparison with the Chernozem in this work), the differences, if any, might be difficult to reveal.
Naumova, N.B.; Belanov, I.P.; Alikina, T.Y.; Kabilov, M.R. Undisturbed Soil Pedon under Birch Forest: Characterization of Microbiome in Genetic Horizons. Soil Syst. 2021, 5, 14. https://doi.org/10.3390/soilsystems5010014
|
|
Point 4 |
L82 106 414 [Error! Bookmark not defined.] |
|
Response 4 |
Corrected.
|
|
Point 5 |
. In the discussion, I would like the authors to note whether there are fundamental differences in biodiversity between soils from West Siberian chernozem and chernozems from other regions.
|
|
Response 5 |
We believe that one time point and one area are not sufficient to draw conclusions about the fundamental differences between the West Siberian chernozem and chernozems from other regions. There are few reports about the microbiome in chernozems. |
|
Point 6 |
Have the elemental composition of soils been analyzed? It would be interesting to see if there is a correlation between the diversity of microbiomes (especially the contribution of actinomycetes) and the presence of biophilic elements and trace elements. |
|
Response 6 |
No, we did not analyze the elemental composition as within the context of the study we did not expect any correlation: based on our experience, albeit limited, of course, we can assure you that, if the soil is exactly of the same genesis, the sampling plots adjacent, and samples taken from the same horizon/layer (this is important!), then one does not find any statistically significant correlation between bacteriobiome and soil properties, including chemical elements. |

Round 2
Reviewer 4 Report
The authors made changes to the manuscript and responded to my comments. I believe that the manuscript can be published without further laughing